# Zero-Shot Image Compression with Diffusion-Based Posterior Sampling

## Abstract

Diffusion models dominate the field of image generation, however they have yet to make major breakthroughs in the field of image compression. Indeed, while pre-trained diffusion models have been successfully adapted to a wide variety of downstream tasks, existing work in diffusion-based image compression require task specific model training, which can be both cumbersome and limiting. This work addresses this gap by harnessing the image prior learned by existing pre-trained diffusion models for solving the task of lossy image compression. This enables the use of the wide variety of publicly-available models, and avoids the need for training or fine-tuning. Our method, PSC (Posterior Sampling-based Compression), utilizes zero-shot diffusion-based posterior samplers. It does so through a novel sequential process inspired by the active acquisition technique "Adasense" to accumulate informative measurements of the image. This strategy minimizes uncertainty in the reconstructed image and allows for construction of an image-adaptive transform coordinated between both the encoder and decoder. PSC offers a progressive compression scheme that is both practical and simple to implement. Despite minimal tuning, and a simple quantization and entropy coding, PSC achieves competitive results compared to established methods, paving the way for further exploration of pre-trained diffusion models and posterior samplers for image compression.

## 1 Introduction

Diffusion models excel at generating high-fidelity images (Ho et al., 2020; Sohl-Dickstein et al., 2015; Song et al., 2020; Dhariwal & Nichol, 2021; Vahdat et al., 2021; Rombach et al., 2022). As such, these models have been harnessed for solving a wide variety of tasks, including inverse problems (Saharia et al., 2021; 2022; Chung et al., 2023; Kawar et al., 2021; 2022a; Song et al., 2023), image editing (Meng et al., 2021; Brooks et al., 2023; Kawar et al., 2023; Huberman-Spiegelglas et al., 2023), and uncertainty quantification (Belhasin et al., 2023). Conveniently, it has been demonstrated that many of these downstream tasks can be solved with a pre-trained diffusion model, thus alleviating the need for task specific training.

Image compression is crucial for efficiently storing and transmitting visual data. This task has therefore attracted significant attention over the past several decades. The core idea in designing an effective compression scheme is to preserve as much of the information in the image while discarding less important portions, resulting in a lossy compression paradigm that introduces a trade-off between image quality and file size. Traditional compression methods, such as JPEG (Wallace, 1991) and JPEG2000 (Skodras et al., 2001), achieve this goal by applying a fixed whitening transform on the image and quantizing the obtained transform coefficients. These algorithms allocate bits dynamically to the coefficients based on their importance, and wrap this process with entropy coding for further lossless compression. More recently, neural compression methods have demonstrated improved performance over their classical counterparts. These techniques employ deep learning and incorporate the quantization and entropy-coding directly into the training loss (Ballé et al., 2018; Minnen et al., 2018; Cheng et al., 2020; Ballé et al., 2016; Theis et al., 2017; Toderici et al., 2015). In this context, deep generative models, such as GANs (Mentzer et al., 2020) or diffusion models (Yang & Mandt, 2024), can be used to improve the perceptual quality of decompressed images, fixing visual artifacts that are commonplace in many classic compression methods, such as JPEG (Wallace, 1991).

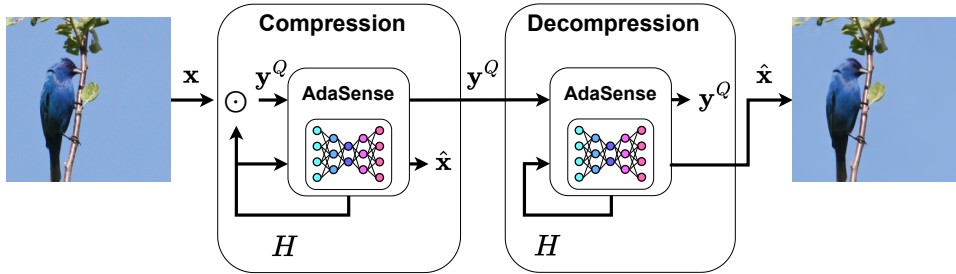

Figure 1: **PSC diagram:** Both the compression and the decompression parts employ the AdaSense algorithm for building an image-specific sensing matrix $H$, to which rows are added progressively based on posterior sample covariance. While the encoder requires access to the real image $\mathbf{x}$ for computing the measurements $\mathbf{y}$, both the encoder and the decoder use the quantized measurements for the AdaSense computations. This, along with a coordinated random seed, guarantee that both sides produce the same deterministic outputs, alleviating the need for transmitting the sensing matrix as side information.

Several works attempted to harness diffusion models for image compression. Many of these methods utilize an existing compression algorithm for the initial compression stage, and use a diffusion model for post-hoc decoding. A notable example for this approach is the family of diffusion-based algorithms for JPEG decoding (Kawar et al., 2022b; Saharia et al., 2022; Song et al., 2023; Ghouse et al., 2023). While these methods show promising results, they remain limited by the inherent shortcomings of the base compression algorithm on which they build. Another approach interleaves training the neural compression component with the diffusion model-based decoding (Careil et al., 2023; Yang & Mandt, 2024; Relic et al., 2024). Such methods reach impressive results, but they require training a task-specific diffusion model and thus cannot exploit the strong prior embedded within large pre-trained models.

In this paper, we introduce PSC (Posterior Sampling-based Compression), a zero-shot image compression method that leverages the general-purpose image prior learned by pre-trained diffusion models. PSC enables exploiting the vast array of publicly available models without requiring training a model that is specific for the compression task. PSC employs a progressive sampling strategy inspired by the recent adaptive compressed sensing method *AdaSense* (Elata et al., 2024). Specifically, in each step PSC utilizes a diffusion-based zero-shot posterior sampler to identify the linear projection of the image that minimizes the reconstruction error. These projections are constructed progressively at the encoder and quantized to form the compressed code. At the decoder side, the exact same calculations are applied (fixing the seed), so that both the encoder and the decoder reproduce exactly the same image-adaptive transform, eliminating the need for transmitting side-information beyond the projections.

We evaluate the effectiveness of PSC on a diverse set of images from the ImageNet dataset (Deng et al., 2009). We compare PSC to established compression methods like JPEG (Wallace, 1991), BPG (Bellard, 2018), and HiFiC (Mentzer et al., 2020) in terms of distortion (PSNR) and image quality. Our experiments demonstrate that PSC achieves superior performance, offering the flexibility to prioritize either low distortion or high image quality (Blau & Michaeli, 2019) based on user preference, all while using the same compressed representation. Furthermore, we explore the potential of using Text-to-Image latent diffusion models (Rombach et al., 2022) for image compression. This approach enables the use of more efficient DNN architectures and incorporates a textual description of the image for better compression. Our Latent-PSC exhibits superior compression results in term of image quality and semantic similarity, suggesting its potential for tasks where preserving image content and meaning is crucial. These experiments showcase the promising results of PSC and its variants, highlighting the potential of pre-trained diffusion models and posterior sampling for efficient image compression.

In summary, the proposed compression approach is a novel strategy that relies on the availability of an (approximate) posterior sampler. The compression is obtained by constructing a sequentially growing image-adaptive transform that best fits the intermediate uncertainties throughout the process.

---

**Algorithm 1** A single iterative step of AdaSense – Denoted as AdaSenseStep $(\boldsymbol{H}_{0:k}, \mathbf{y}_{0:k}, r)$

---

**Require:** Previous sensing rows $\boldsymbol{H}_{0:k}$, corresponding measurements $\mathbf{y}_{0:k}$, number of new measurements $r$

1: $\{\mathbf{x}_i\}_{i=1}^s \sim p_{\mathbf{x}|\boldsymbol{H}_{0:k}, \mathbf{y}_{0:k}}$                     $\triangleright$ generate $s$ posterior samples
2: $\{\mathbf{x}_i\}_{i=1}^s \leftarrow \{\mathbf{x}_i - \frac{1}{s}\sum_{j=1}^s \mathbf{x}_j\}_{i=1}^s$                $\triangleright$ center samples
3: $\tilde{\boldsymbol{H}} \leftarrow$ Append top $r$ right singular vectors of $(\mathbf{x}_1, \ldots, \mathbf{x}_s)^\top$     $\triangleright$ select $r$ principal components
4: **return** $\tilde{\boldsymbol{H}}$

---

This work presents an initial exploration that employs a simplified quantization strategy, and lacks tailored entropy coding. Also, the proposed approach incurs a high computational cost. Nevertheless, we believe that the presented method represents a promising direction for future research. Advancements in diffusion-based posterior samplers and our proposed training-free compression scheme have the potential to lead to significant improvements in compression of images or other signals of interest.

## 2 BACKGROUND

Our proposed compression scheme, PSC, leverages AdaSense (Elata et al., 2024), a sequential adaptive compressed sensing algorithm that gathers optimized linear measurements that best represent the incoming image. Formally, for inverse problems of the form $\mathbf{y} = \boldsymbol{H}\mathbf{x}$ with a sensing matrix $\boldsymbol{H} \in \mathbb{R}^{d \times D}$ ($d < D$), we would like to select $\boldsymbol{H}$ for reconstructing a signal $\mathbf{x} \in \mathbb{R}^D$ from the linear measurements $\mathbf{y} \in \mathbb{R}^d$ with a minimal possible error. AdaSense starts with an empty matrix and selects the rows of $\boldsymbol{H}$ sequentially. At stage $k$, we have the currently held[1] matrix $\boldsymbol{H}_{0:k}$ and measurements $\mathbf{y}_{0:k} = \boldsymbol{H}_{0:k}\mathbf{x}$. The selection of the next row is done by generating samples from the posterior $p(\mathbf{x}|\boldsymbol{H}_{0:k}, \mathbf{y}_{0:k})$ using some zero-shot diffusion-based posterior sampler (Kawar et al., 2022a; Chung et al., 2023; Manor & Michaeli, 2023; Song et al., 2023). The posterior samples are used to identify the principal direction of uncertainty, defined as the MMSE estimation error, via PCA. This direction is chosen as the next row in $\boldsymbol{H}$, which is used to acquire a new measurement of $\mathbf{x}$. More generally, instead of selecting one new measurement, it is possible to add $r$ new measurements in each iteration. A single iteration of AdaSense is described in Algorithm 1, and should be repeated $d$ times. This algorithm presents a strategy of choosing the $r$ leading eigenvectors of the PCA at every stage instead of a single one, getting a substantial speedup in the measurements' collection process at a minimal cost to adaptability.

AdaSense relies on the availability of a posterior sampling method, which can be chosen according to the merits and pitfalls of existing samplers. Using a zero-shot diffusion-based posterior sampler (Kawar et al., 2022a; 2021; Song et al., 2023; Chung et al., 2023; 2022) enables the use of one of the many existing pre-trained diffusion models. The described process produces an image-specific sensing matrix $\boldsymbol{H}$ and corresponding measurements $\mathbf{y}$, and these can be used for obtaining a candidate reconstruction $\hat{\mathbf{x}}$ by leveraging the final posterior, $p(\mathbf{x}|\boldsymbol{H}, \mathbf{y})$, where $\boldsymbol{H}$ is the final matrix (obtained at the last step). This final reconstruction step can lean on a different posterior sampler, more adequate for this task (e.g., choosing a slower yet more exact method, while relying on the fact that it is applied only once). Please refer to the original publication for derivations.

## 3 PSC: THE PROPOSED COMPRESSION METHOD

We start by describing the commonly used transform-based compression paradigm, as practiced by classical methods, and then contrast this with PSC – our proposed approach. Image compression algorithms, like JPEG (Wallace, 1991), apply a pre-chosen, fixed and Orthonormal[2] transform on the input image, $\mathbf{x} \in \mathbb{R}^D$, obtaining its representation coefficients. These coefficients go through a quantization stage, in which portions of the transform coefficients are discarded entirely, and other portions are replaced by their finite representation, with a bit-allocation that depends on their im-

---

[1]In our notations, the subscript $\{0 : k\}$ implies that $k$ elements are available, from index 0 to index $k - 1$.

[2]Having orthogonal rows has two desirable effects – easy-inversion and a whitening effect. Using a biorthogonal system as in JPEG2000 (Skodras et al., 2001) has similar benefits.

---

**Algorithm 2** PSC: Posterior Sampling Compression

---

**Require:** Image $\mathbf{x}$, number of steps $N$, number of measurements per step $r$.
1: **if** Encoder **then initialize** $\mathbf{y}_{0:0}$ as an empty vector
2: **else** Decoder **then initialize** $\mathbf{y}_{0:Nr}$ from compressed representation
3: **for** $n \in \{0 : N-1\}$ **do**
4:     $\boldsymbol{H}_{nr:nr+r} \leftarrow \text{AdaSenseStep}\left(\boldsymbol{H}_{0:nr}, \mathbf{y}_{0:nr}, r\right)$   ▷ use Algorithm 1 to obtain the next $r$ rows
5:     **if** Encoder **then**
6:         $\mathbf{y}_{0:nr+r} \leftarrow \text{Append}\left[\mathbf{y}_{0:nr}, Q(\boldsymbol{H}_{nr:nr+r}\mathbf{x})\right]$   ▷ measure the real image $\mathbf{x}$ and quantize
7:     **else** Decoder **then**
8:         $\mathbf{y}_{0:nr+r} \leftarrow \text{Append}\left[\mathbf{y}_{0:nr}, \mathbf{y}_{nr:nr+r}\right]$   ▷ measurements from compressed representation
9:     $\boldsymbol{H}_{0:nr+r} \leftarrow \text{Append}\left[\boldsymbol{H}_{0:nr}, \boldsymbol{H}_{nr:nr+r}\right]$
10: **return** $\mathbf{x}_1 = f(\mathbf{y}_{0:Nr}, \boldsymbol{H}_{0:Nr})$   ▷ posterior sampling or alternative restoration

---

portance for the image being compressed. As some of the transform coefficients are discarded, this scheme can be effectively described as using a partial transform matrix $\boldsymbol{H} \in \mathbb{R}^{d \times D}$ with orthogonal rows, and applying the quantization function $Q(\cdot)$ to the remaining measurements $\mathbf{y} = \boldsymbol{H}\mathbf{x}$. Under the assumption that the obtained coefficients are (nearly) statistically independent, the quantization may operate scalar-wise on the elements of $\mathbf{y}$ effectively. Image compression algorithms include an entropy coding stage that takes the created bit-stream and passes it through a lossless coding block (e.g. Huffman coding, arithmetic coding, etc.) for a further gain in the resulting file-size. Just to complete the above description, the decoder has knowledge of the transform used, $\boldsymbol{H}$; it obtains $Q(\mathbf{y})$ and produces the image $\boldsymbol{H}^{\dagger}Q(\mathbf{y})$ as the decompressed output.

When an algorithm is said to be progressive, this means that the elements of $\mathbf{y}$ are sorted based on their importance, and transmitted in their quantized form sequentially, enabling a decompression of the image at any stage based on the received coefficients so far. Progressive compression algorithms are highly desirable, since they induce a low latency in decompressing the image. Note that the progressive strategy effectively implies that the rows of $\boldsymbol{H}$ have been sorted as well based on their importance, as each row gives birth to the corresponding element in $\mathbf{y}$. Adopting this view, at step $k$ we consider the sorted portions of $\boldsymbol{H}$ and $\mathbf{y}$, denoted by $\boldsymbol{H}_{0:k} \in \mathbb{R}^{k \times D}$ and $\mathbf{y}_{0:k} = \boldsymbol{H}_{0:k}\mathbf{x} \in \mathbb{R}^{k}$. As the decoder gets $Q(\mathbf{y}_{0:k})$, it may produce $\boldsymbol{H}_{0:k}^{\dagger}Q(\mathbf{y}_{0:k})$ as a temporary output image.

In this work we propose PSC (Posterior Sampling-based Compression) – a novel and highly effective lossy compression scheme. PSC shares much with the above description: A linear orthogonal transform is applied, a scalar-wise quantization of the coefficients is deployed, an entropy coding stage is used as well, and the overall structure of PSC is progressive. However, the major difference lies in the identity of $\boldsymbol{H}$: Rather than choosing $\boldsymbol{H}$ to be a fixed matrix, PSC constructs it row-by-row, while fully adapting it's content with the incoming image to be compressed. This modus-operandi is counter-intuitive, as the immediate question that comes to mind is this: How would the decoder know which transform to apply in recovering the image? PSC answers this question by leaning on the progressive compression structure adopted. The core idea is to use the currently held matrix $\boldsymbol{H}_{0:k}$ and the quantized measurements $Q(\mathbf{y}_{0:k}) \in \mathbb{R}^{k}$, available in both the encoder and the decoder, for computing the next row, $\mathbf{h}_k \in \mathbb{R}^{1 \times D}$, identically on both sides. This row joins the matrix $\boldsymbol{H}_{0:k}$, obtaining the transform matrix for the next step,

$$\boldsymbol{H}_{0:k+1} = \begin{bmatrix} \boldsymbol{H}_{0:k} \\ \mathbf{h}_k \end{bmatrix} \in \mathbb{R}^{(k+1) \times D}. \tag{1}$$

Once created, the encoder projects the image onto the new direction, $y_k = \mathbf{h}\mathbf{x}$, and a quantized version of this value is transmitted to the decoder.

Clearly, the key for the above process to operate well is the creation of $\mathbf{h}_k$ based on the knowledge of $Q(\mathbf{y}_{0:k})$. This is exactly where AdaSense comes into play. PSC's compression algorithm leverages the AdaSense scheme (described in Section 2) to generate the same sensing matrix $\boldsymbol{H}$ in the encoder and the decoder, thereby avoiding the need for side-information. Specifically, the encoder and decoder algorithms share the same seeds, the same accumulated matrix $\boldsymbol{H}_{0:k}$ and the same measurements $Q(\mathbf{y}_{0:k})$, ensuring the next row of the sensing matrix $\boldsymbol{H}$ is identical on both sides. Interestingly, as a by-product of the AdaSense algorithm, the obtained sensing matrix $\boldsymbol{H}$ has orthogonal rows, disentangling the measurements, as expected from a compression algorithm.

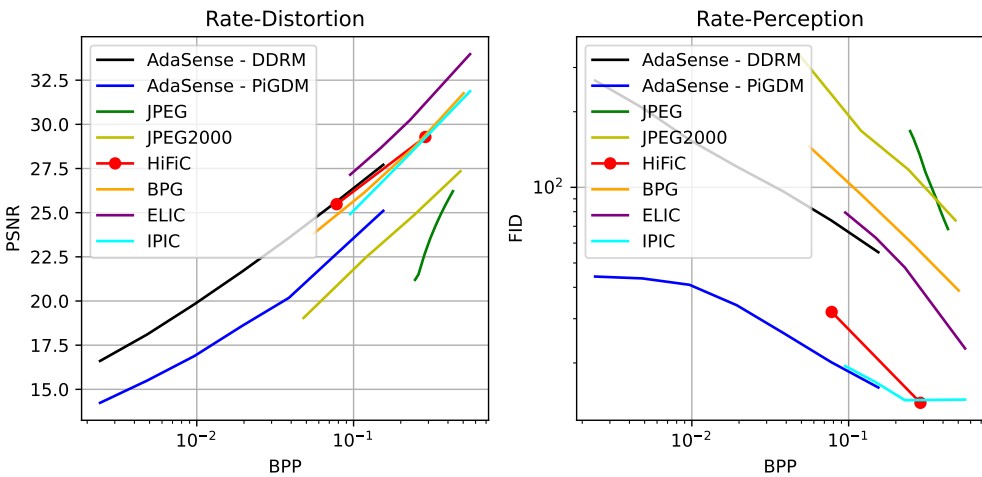

Figure 2: **Rate-Distortion (left) and Rate-Perception (right) curves for ImageNet256 compression.** Distortion is measured as average PSNR of images for the same desired rate or specified compression quality, while Perception (image quality) is measured by FID.

For completeness of our disposition, here is a more detailed description of the AdaSense/PSC computational process for evaluating $\mathbf{h}_k$. Consider the posterior probability density function $p(\mathbf{x}|\boldsymbol{H}_{0:k}, Q(\mathbf{y}_{0:k}))$. This conditional PDF describes the probability of all images that comply with the accumulated measurements so far. By evaluating the first two moments of this distribution, $\mu_k \in \mathbb{R}^D$ and $\Sigma_k \in \mathbb{R}^{D \times D}$, we get access to the spread of these images. Notice that the original image being compressed, $\mathbf{x}$, is likely to reside within the area of high probability of this conditional Gaussian. Thus, by choosing $\mathbf{h}_k$ to be the eigenvector corresponding to the largest eigenvalue of $\Sigma_k \in \mathbb{R}^{D \times D}$, we get a highly informative direction on which to project $\mathbf{x}$, so as to get the most valuable incremental information about it. Knowing $y_k = \mathbf{h}_k \mathbf{x}$ (or its quantized value) implies that we have reduced the uncertainty of the candidate images probable in the posterior distribution $p(\mathbf{x}|\boldsymbol{H}_{0:k}, Q(\mathbf{y}_{0:k}))$ in the most effective way. PSC (like AdaSense) deploys a diffusion-based posterior sampler that can handle inverse problems of the form[3] $\mathbf{y} = \boldsymbol{H}\mathbf{x}$, enabling the use of publicly available pre-trained diffusion models, without any additional training. By drawing many such samples from the posterior, we can compute their PCA, which provides a reliable estimate of the top principal component of the true posterior covariance.

The detailed procedures for compression and decompression with PSC are presented in Algorithm 2. Here as well we consider a possibility of working with blocks of $r$ measurements at a time for speed-up consideration. A diagram of our proposed method is provided in Figure 1, and a comprehensive pseudo-code implementation is included in Appendix C. In our implementation we focus on a simple quantization approach, reducing the precision of $\mathbf{y}$ from float32 to float8 (Micikevicius et al., 2022). We employ Range Encoding implemented using (Bamler, 2022) as an entropy coding on the quantized measurements. The quantization, the posterior sampler and the entropy coding could all be improved, posing promising directions for future work. Finally, after reproducing $\boldsymbol{H}$ on the decoder side, PSC can leverage a (possibly different) posterior sampler to produce the decompressed output $\hat{\mathbf{x}}$.

To summarize, PSC facilitates a greedy step-wise optimal decrease in the volume of the posterior by the accumulated directions chosen, and the corresponding measurements computed with them. This way, the overall manifold of high quality images is intersected again and again, narrowing the remaining portion, while zeroing on the given image $\mathbf{x}$. The progressive nature of PSC provides a key advantage in its flexibility. The same compression algorithm can be used to achieve different points in the Rate-Distortion-Perception (RDP) trade-off space (Blau & Michaeli, 2019). Lower compression rates can be achieved by using fewer measurement elements, potentially increasing the perceived distortion. Note that, just like AdaSense, PSC may use a different final posterior

---

[3]In sampling from the Posterior, we disregard the quantization, thus resorting to approximate samplers.

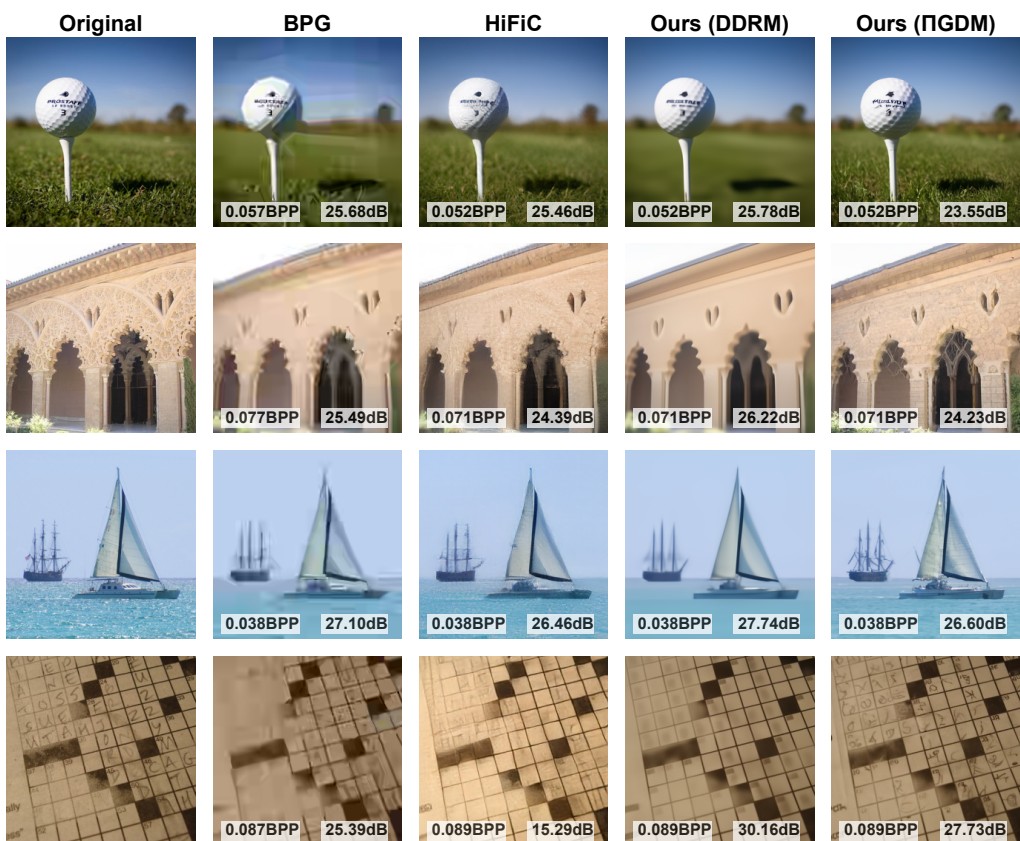

Figure 3: **Qualitative examples for compression with PSC, compared to other compression algorithms with similar BPP.** BPP and PSNR are reported per each. Our method can be used for both low-distortion with DDRM or high perceptual quality with ΠGDM using the same compressed representation.

sampler during decompression, in an attempt to further boost perceptual quality for the very same measurements. In contrast to all the above, many other compression methods using generative models, e.g., HiFiC (Mentzer et al., 2020; Careil et al., 2023; Yang & Mandt, 2024), require separate training of both the encoder and decoder changing the rate or traversing the RDP function. This fixed configuration limits their ability to adapt to different compression demands.

## 4  EXPERIMENTS

We evaluate the performance of PSC on color images from the ImageNet (Deng et al., 2009) dataset. We compare distortion (PSNR) and bit-per-pixel (BPP) averaged on a subset of validation images, using one image from each of the 1000 classes, following (Pan et al., 2021). Unconditional diffusion models from (Dhariwal & Nichol, 2021) are used for images of size $256 \times 256$. We apply Algorithm 1 to progressively decode at higher rates, selecting $r = 12$ and using $s = 16$ posterior samples with 20 DDRM steps, as detailed in Appendix A. The choice of hyperparameter is accounted for in Appendix B.

A key advantage of PSC is its ability to prioritize perceptual quality during decompression by changing the final reconstruction algorithm. However, this flexibility comes with a caveat: using a high-quality reconstruction algorithm will inevitably lead to higher distortion (Blau & Michaeli, 2019). Despite this, using PSC, the same compressed representation can be decoded using either a low-distortion or high perceptual quality approach with minimal additional computational cost. Specifically, we find that ΠGDM (Song et al., 2023) produces the highest quality images for our reconstruction problem, while DDRM (Kawar et al., 2022a) leads to the lowest distortion.

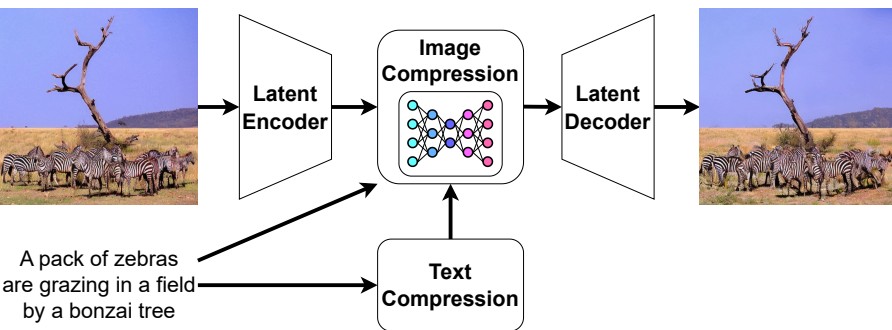

Figure 4: **Latent-PSC diagram:** Latent Text-to-Image diffusion models such as Stable Diffusion can be used for effective image compression with PSC. The latent representation is compressed using linear measurements. The textual prompt is used for conditioning the diffusion model in both the compression and decompression, and thus this text is also transmitted.

Figure 2 presents the rate-distortion and rate-perception curves of PSC compared to several established methods: classic compression techniques like JPEG (Wallace, 1991), JPEG2000 (Skodras et al., 2001), and BPG (Bellard, 2018). We also compare to neural compression methods, such as ELIC (He et al., 2022) and its diffusion-based derivative IPIC (Xu et al., 2024), as well as HiFiC (Mentzer et al., 2020), a prominent GAN-based neural compression method. Distortion is measured by averaging the PSNR across different algorithms for a given compression rate. Image quality is quantified using FID (Heusel et al., 2017), estimated on 50 random $128 \times 128$ crops from each image, compared to the same set of baselines. The graphs demonstrate that PSC achieves comparable or superior performance, particularly at low BPP regimes, when considering both distortion and image quality. Figure 3 showcases qualitative image samples compressed using different algorithms at the same rate, further supporting our findings. Notably, PSC achieves exceptional image quality despite the fact that it does not require any task-specific training for compression.

Latent Text-to-Image diffusion models have gained popularity due to their ease-of-use and low computational requirements. These models employ a VAE (Kingma & Welling, 2013) to conduct the diffusion process in a lower-dimensional latent space (Vahdat et al., 2021; Rombach et al., 2022). In this work we also explore the integration of PSC with Stable Diffusion (Rombach et al., 2022), a publicly available latent Text-to-Image diffusion model. This variant, named Latent-PSC, operates in the latent space of the diffusion model. Both compression and decompression occur within this latent space, leveraging the model's VAE decoder to reconstruct the image from the decompressed latent representation. Additionally, we condition all posterior sampling steps on a textual description, which must be given along with the original image or inferred using an image captioning module (Vinyals et al., 2016; Li et al., 2022; 2023). The text prompt must be added to the compressed representation to avoid side-information. A detailed diagram of Latent-PSC is presented in Figure 4.

We evaluate Latent-PSC on $512 \times 512$ images from the MSCOCO (Lin et al., 2014) dataset, which includes textual descriptions for each image. We compress the textual description assuming 6 bits per character, with no entropy encoding. Figure 5 shows decompressed samples using Latent-PSC with different rates, demonstrating good semantic similarity to the originals and high perceptual quality. While Latent-PSC exhibits promising results, we observe a significant drop in PSNR when decoding the images using the VAE decoder. This is not unexpected, as simply encoding and decoding images without compression also leads to a noticeable PSNR reduction. We believe that future advancements in latent-to-pixel-space decoding methods have the potential to address this limitation.

Figure 6 illustrates the impact of using a captioning model to obtain the textual representation. In this experiment, the captions generated by BLIP (Li et al., 2022) achieved comparable or superior results to human annotated description from the dataset. However, omitting the prompt causes some degradation of quality.

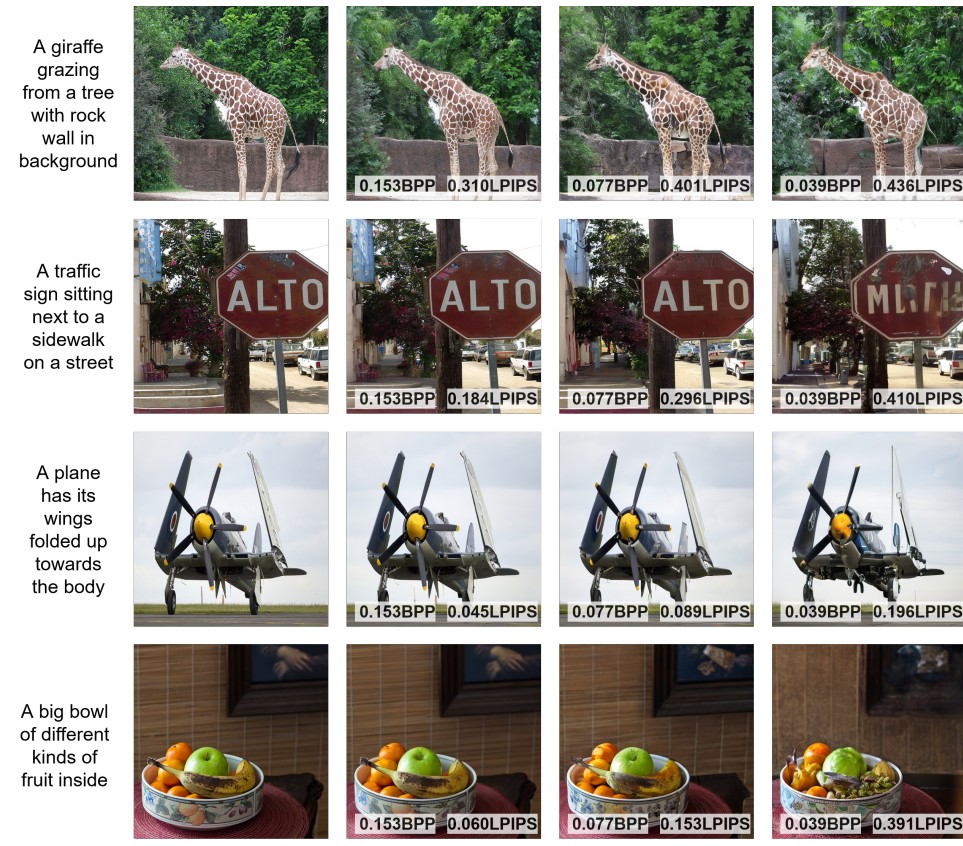

Figure 5: **Qualitative examples of Latent-PSC with Stable Diffusion.** For each image and corresponding text, several results for different bit-rates are shown. BPP and LPIPS are reported.

## 5 RELATED WORK

Diffusion models have been used in tandem with existing classical compression algorithms, providing an alternative data-driven decompression scheme for high-perceptual quality reconstruction (Ghouse et al., 2023; Saharia et al., 2022). Among those, several works attempt to preform zero-shot diffusion based reconstruction (Kawar et al., 2022b; Song et al., 2023), creating a training-free decompression method. Unlike our proposed approach, these works are limited to specific compression algorithms, which may be lacking. A recent work by Xu et al. (2024) attempts to utilize general diffusion-based posterior samplers to decode a compressed representation created with a neural compression method into a high-quality image. While this methods uses pre-trained diffusion-based posterior samplers similar to our method, it differs in its goal to traversing the RDP trade-off (Blau & Michaeli, 2019) of existing neural compression schemes.

Recent advancements combine neural compression for the encoding stage and diffusion models for decompression. The straightforward approach uses separate (Hoogeboom et al., 2023) or joint (Yang & Mandt, 2024) neural compression and diffusion training to create a compact compressed representation, and a conditional diffusion model for decompressing this representation into high-quality images. A similar approach is taken by (Careil et al., 2023; Relic et al., 2024), which makes use of latent diffusion (Rombach et al., 2022) and text-conditioned models to make training more simple and efficient. While promising, these methods require complex rate-specific training for compression, hindering their flexibility. Works such as Gao et al. (2022) tackle this issue and offer a method for training-free post-hoc reconfiguration of a neural-compression model's rate, yet at the cost of high computational cost and drop to performance. Similarly,

Interestingly, the concept of using pre-trained diffusion models for compression was initially introduced in the DDPM publication (Ho et al., 2020). However, their proposed approach focused

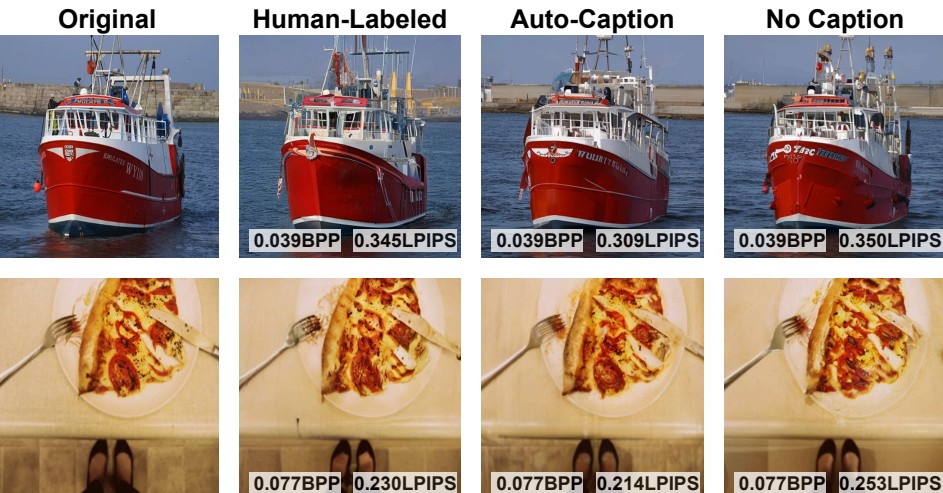

Figure 6: **Qualitative examples of Latent-PSC with various prompt configurations.** For each image we compare compression results with human annotated textual description, auto-captioning using a model, and using no caption.

on the theoretical compression limit and did not propose a practical compression algorithm. Theis et al. (2022) analyzes a similar theoretical limit based on a more realistic reverse channel coding techniques (Li & El Gamal, 2018). However, their implementation suffers from high computational complexity and lacks publicly available code, preventing a direct comparison with our approach.

## 6    LIMITATIONS AND DISCUSSION

While PSC offers a novel perspective on using generative models for compression, it remains a preliminary study with several limitations. The primary limitation is PCS's high computational cost, caused by the recurring sampling using a diffusion model. Thus, our algorithm typically requires approximately $10,000$ NFEs, depending on the desired rate. PSC's reliance on posterior sampling also inherently ties the capabilities of our method to the quality of zero-shot posterior sampler. Fortunately, there is ongoing research focused on improving the speed and quality of diffusion models and posterior sampling, which could significantly reduce this limitation in the future. The current implementation utilizes an oversimplified quantization strategy for the measurements. Employing a more sophisticated quantization method has the potential to significantly improve compression rates. Exploring advanced quantization techniques is a promising avenue for future research. Lastly, PSC is currently limited to linear measurements due to the capabilities of existing posterior samplers, as well as the complexity of optimizing non-linear measurements. Investigating the use of non-linear measurements along with corresponding inverse problem solvers could potentially lead to further improvements in compression performance.

## 7    CONCLUSION

This work introduces PSC, a novel zero-shot diffusion-based image compression method. PSC utilizes a posterior sampler to progressively acquire informative measurements of an image, forming a compressed representation. The decompression reproduces the steps taken in the compression algorithm using the encoded measurements, to finally reconstruct the desired image. PSC is simple to implement, requires no training data, and demonstrates flexibility across various image domains. We believe that future progress would offer better quantization algorithms along with matching sampling procedures, and lead to a further improvement in image compression.

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
