## A   IMPLEMENTATION DETAILS

For the selection of the image-specific sensing matrix $H$ we used the unconditional diffusion models from Dhariwal & Nichol (2021) and 25 DDRM (Kawar et al., 2022a) steps to generate 16 samples, using $\eta = 1.0$ and $\eta_b = 0.0$. We added 12 rows to $H$ in every iteration, and used matched the number of iterations to the desired rate. We restore the images using the same model with either $\Pi$GDM (Song et al., 2023) with 100 denoising steps and default hyperparameters for high perceptual quality restoration, or an average of 64 DDRM (Kawar et al., 2022a) samples which where produced as detailed above for low-distortion restoration.

In the latent diffusion experiment we used stable-diffusion-2-base[4] (Rombach et al., 2022) and 25 DDPM steps with projection to generate 64 samples for selecting the sensing matrix. We added 48 rows to $H$ in every iteration. We restore the images using the same model with either $\Pi$GDM (Song et al., 2023) with 100 denoising steps and default hyperparameters. To increase the decoded image's perceptual quality, we do not use the $\Pi$GDM modification to the sampling algorithm in the last 5 steps.

We used publictly available third party software for JPEG (Wallace, 1991), JPEG2000 (Skodras et al., 2001), and BPG (Bellard, 2018). For HiFiC (Mentzer et al., 2020), we trained our own model using the pytorch implementation publicly available on github [5]. We trained the models using the default parameters for each rate, and pruned networks that failed to converge to the desired rate.

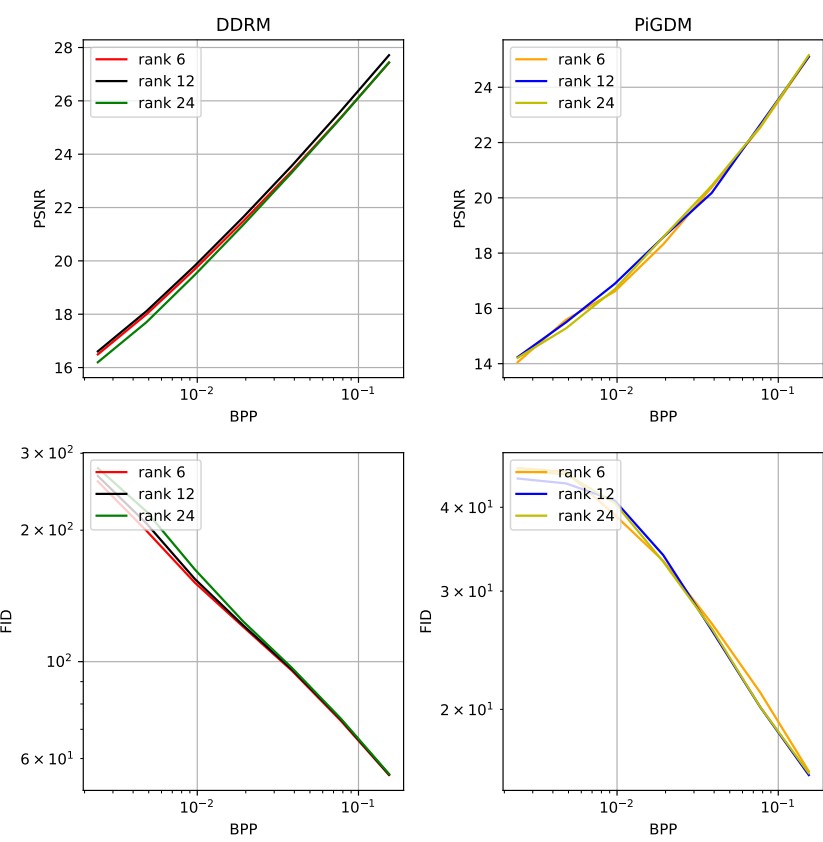

Figure 7: **Rate-Distortion (top) and Rate-Perception (bottom) curves for ImageNet256 compression, using DDRM (left) and PiGDM (right).** Distortion is measured as average PSNR of images for the same desired rate or specified compression quality, while Perception (image quality) is measured by FID.

---

[4]https://huggingface.co/stabilityai/stable-diffusion-2-base
[5]https://github.com/Justin-Tan/high-fidelity-generative-compression

FID (Heusel et al., 2017) is measured using Pytorch Fidelity [6], and the Range Encoder from constriction (Bamler, 2022) as an entropy encoder[7].

## B  EFFECT OF MEASUREMENT RANK

We repeat the imagenet experiment with different values of the hyperparameter $r$, which determines how adaptive our algorithm would be. We modify the number of samples generated at each iteration $s$ accordingly to account for the rank required by the emperical covariance matrix. Based on the original implementation of AdaSense (Elata et al., 2024), we expect performance to improve the lower the value of $r$ is. In the results, demonstrated in Figure 7, the variation of the rank seems to have only a marginal effect, even for low rates. We conclude that PSC is not sensitive to this parameter, and $r$ can be tuned according to the system's hardware (namely, maximum available batch size).

## C  PSC PSEUDO-CODE

```python
from utilities import posterior_sampler, restoration_fuction,
    entropy_encode, entropy_decode

def AdaSense_Step(H, y, r, shape, s=None):
    c, h, w = shape
    s = s if s is not None else (r * 4) // 3
    noise = torch.randn((s, c, h, w))
    samples = posterior_sampler(noise, H, y)
    samples = samples.reshape(s, -1)
    samples = samples - samples.mean(0, keepdim=True)
    new_rows = torch.linalg.svd(samples, full_matrices=False)[-1][:r]
    return new_rows

def PSC_compress(image, N, r)
    c, h, w = image.shape
    H = torch.zeros((0, c * h * w))          # Empty sensing matrix
    y = H @ image.reshape((-1, 1))           # Empty measurements
    compressed_representation = y.clone()

    for n in range(N):
        new_rows = AdaSense_Step(H, y, r, (c, h, w))
        H = torch.cat([H, new_rows])
        y = torch.cat([y, new_rows @ image.reshape((-1, 1)])

        compressed_representation = y.to(torch.float8_e4m3fn) # Quantize
        y = compressed_representation.to(torch.float32)

    return entropy_encode(compressed_representation)

def PSC_decompress(compressed_representation, N, r)
    compressed_representation = entropy_decode(compressed_representation)
    c, h, w = image.shape
    H = torch.zeros((0, c * h * w))          # Empty sensing matrix
    y = H @ image.reshape((-1, 1))           # Empty measurements

    for n in range(N):
        new_rows = AdaSense_Step(H, y, r, (c, h, w))
        H = torch.cat([H, new_rows])
        y = compressed_representation[:(n*r)].to(torch.float32)

    return restoration_fuction(H, y)
```

Our complete code will be published upon acceptance.

---

[6]https://github.com/toshas/torch-fidelity
[7]https://github.com/bamler-lab/constriction

# D    IMAGE SPECIFIC RATE-DISTORTION

Below in Figure 8, we present image-specific rate-distortion curves for the images displayed in
Figure 3. These graphs provide additional evidence that the trends shown in Figure 2 is general to
many images and not only to their avarage.

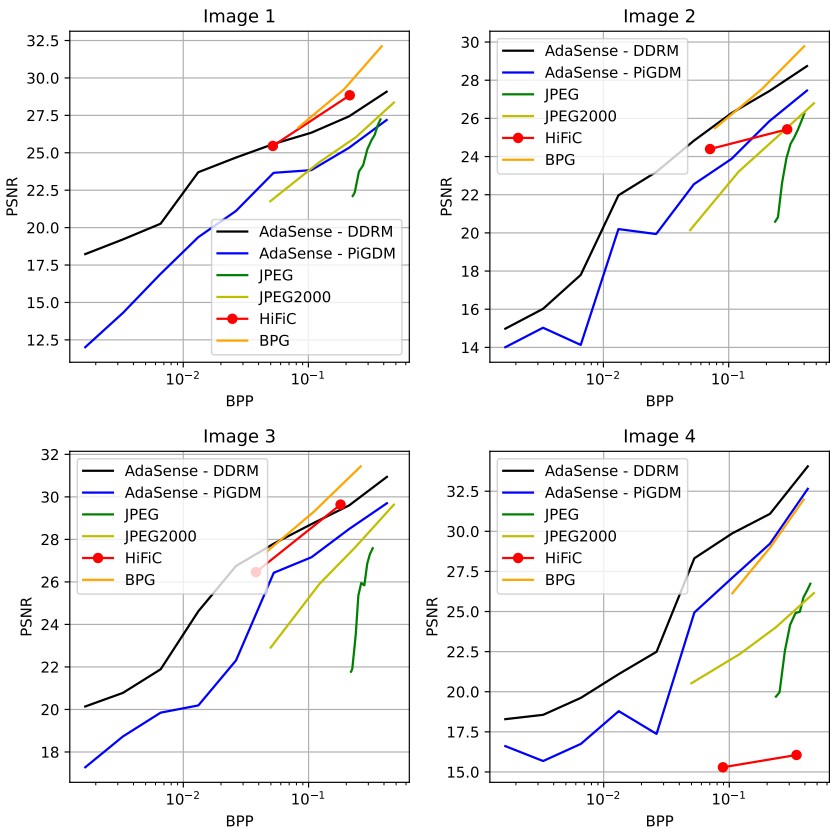

Figure 8: **Rate-Distortion curves for specific images from ImageNet256.** The images from Figure 3 are used, numbered from top to bottom. Distortion is measured by PSNR of images.