# OpenReview forum: "Zero-Shot Image Compression with Diffusion-Based Posterior Sampling"
_ICLR.cc/2025/Conference — ICLR 2025 Conference Withdrawn Submission_

### Official Review · Reviewer_Cfmc · 2024-10-30

**Soundness:** 3
**Presentation:** 3
**Contribution:** 3
**Rating:** 6
**Confidence:** 4

**Summary:**

This paper transplants a compressed sensing approach to neural image compression. It turns out that this compressed sensing approach works quite well. The proposed approach is simple, as the transforms are linear. Besides, no training is required. Further, the result is also promising compared with HiFiC.

**Strengths:**

It is good to know that a linear transform is enough to achieve a high performance codec. Further, this paper is the first zero-shot perceptual codec for images of size 512x512 using latent diffusion models.

**Weaknesses:**

The first thing that confuse me is the impact of variance minimization of Adasense. It seems to me that variance minimization is important in compressed sensing. However, I am not sure about how variance minization will contribute to image codec. For example, is it possible to improve performance by chaning variance minization to entropy minization? As this target seems to be aligned with codec target.

The second thing that confuse me is the simplicity of transformation. Adopting a simple linear transform is great if the authors want to design a light weight codec. However, clearly this codec is not light weighted. As $\Pi$GDM does not limit the transformation to be linear, it becomes confusing why the authors stick to a linear transformation. Perhaps better performance can be achieved with non-linear transformation.

As this paper works on low bitrate regime, it is better to compare with [Towards image compression with perfect realism at ultra-low bitrates]. Further, as this paper works on zero-shot perceptual image compression, it is better to discuss and compare with [Idempotence and Perceptual Image Compression].

**Questions:**

I am curious about why the variance minimizing target works well in terms of rate-distortion performance for image compression.

---

> ### Author Response · Authors · 2024-11-20
>
> - **Variance Minimization:** The reviewer makes an interesting point of directly minimizing the entropy. Yet, finding the best measurement at each step for lowering the entropy of an unknown distribution is not trivial. By defining the uncertainty as the MSE attained by the linear minimum-MSE (MMSE) predictor of x based on the measurements y, the uncertainty directly correlates to the posterior distribution’s covariance. Thus, a closed-form solution for the optimal linear measurement exists. This closed-form solution indirectly reduces the entropy. For example, if the posterior distribution is a Gaussian, the optimal linear measurement for lowering L2 error is also the optimal in reducing the entropy. This is because a Gaussian distribution remains a Gaussian when conditioned on a linear measurement, and its entropy is proportional to the log determinant of the covariance matrix. We will add this discussion in the paper.
>  - **Non-Linear Measurements:** while diffusion-based non-linear solvers (such as PiGDM) exist, they have yet to be employed for an active acquisition scheme (such as AdaSense). This is probably because finding the optimal non-linear measurements even for L2 error is non-trivial, unlike the linear case. Nevertheless, we demonstrate that even when limited to linear transforms, by constructing an image-specific transform we are able to formulate a highly effective compression scheme.
>  - **Related Work:** We will refer to these works in a revision (we already refer to [1]). Yet, we note that [1] and [2] both use networks that are trained specifically for the task of image compression, unlike our method, which is entirely based on off-the-shelf pre-trained generative networks. Additionally, our compression scheme is independent from neural compression methods, unlike the cited works which integrate existing neural compression methods with advances in conditional image generation, thus providing a novel take on the compression task.
>
> [1] Careil, Marlene, et al. "Towards image compression with perfect realism at ultra-low bitrates." The Twelfth International Conference on Learning Representations. 2023.
>
> [2] Xu, Tongda, et al. "Idempotence and perceptual image compression." arXiv preprint arXiv:2401.08920 (2024).

---

> > ### Comment · Reviewer_Cfmc · 2024-11-26
> >
> > Thanks for the rebuttal. I still recommend to accept this paper. It is interesting to see a zero-shot codec for latent diffusion and 512x512 wild images.

---

### Official Review · Reviewer_T16E · 2024-11-02

**Soundness:** 3
**Presentation:** 2
**Contribution:** 2
**Rating:** 5
**Confidence:** 3

**Summary:**

The paper introduces a new compression algorithm leveraging pre-trained diffusion models combined with AdaSense, a sequential adaptive compressed sensing algorithm that selects the most relevant eigenvectors obtained through PCA to restore compressed data. The results are compared with classical compressors like JPEG, JPEG2000, BPG, and a GAN-based method, HiFiC.

**Strengths:**

The first half of the paper is well-written and well-structured.

The proposed algorithm compresses and decompresses an image, producing a result closer to the original compared to other methods. Compressing based on data importance seems interesting to me.

The key idea behind their algorithm is between lines 184 and 191, and they achieved this progressive decompression using the AdaSense algorithm on the held matrix H.

 Indeed, the question, "How would the decoder know which transform to apply in recovering the image?" in line 198 was on my mind throughout the reading. The narrative of the paper is clear enough to answer such questions directly during the reading.

**Weaknesses:**

Line 466 states, "The primary limitation with our proposed method is its high computational cost." However, there is no discussion of computational time in the paper. JPEG, for example, may have lower quality but offers low latency. Timing is essential in compression/decompression algorithms, yet this information has been omitted in the paper.

The pre-trained diffusion model is a minor part of the algorithm. It’s not even the most relevant component, and yet the authors emphasize the zero-shot diffusion element in the title and abstract, while the main focus of the paper is on earlier steps, using diffusion only for the final posterior sampling. I would be interested in the authors' opinion on this emphasis.

While the mathematical description in the first half of the paper is accurate, I find the experimental section somewhat superficial.

In Fig. 2, it is unclear what the lines represent—I would expect a single point per dataset. Additionally, Fig. 2 is not well described in the text. The same applies to Fig. 4, which lacks detail on the method. It is presented as a high-level figure without adequate explanation in the text. Similarly, in Fig. 5, it’s unclear why the results degrade significantly in the columns on the right—perhaps it’s just a different seed? The same question applies to Fig. 6.

**Questions:**

Look at weaknesses.

---

> ### Author Response · Authors · 2024-11-20
>
> - **Computational Costs:** As mentioned, PSC requires many NFEs to generate the multitudes of samples required at each step. More specifically, we used an order of 10,000 NFEs for compressing each image (depending on the required rate of course, as our method is progressive). While this computational cost is much higher than JPEG, so is the image quality while also maintaining much lower distortion for the same rates. We stress that this is the first introduction of such a posterior-sampler based compression method, and we expect that major speed-ups could be made in the near future, with the improvements of diffusion models and posterior samplers (such as flow-based methods) in general, and with specific algorithmic ideas that fit the architecture of the proposed algorithm.
>  - **Pre-trained Diffusion Model:** The diffusion model is central to the proposed compression algorithm: It is used for the active acquisition phase of the compression as well as for the final restoration step. This model provides the image prior for the compression algorithm by guiding the measurement process. At each step, a multitude of posterior samples are generated with the pretrained diffusion using a posterior sampler, enabling the selection of the next measurement. For these reasons, we believe the emphasis on the zero-shot diffusion to be appropriate.
>  - **Experimental Section:** Could the reviewer please highlight what experiments would benefit the paper? We will gladly add more experiments.
>  - **Figures:** We will make an effort to expand on the figures in the text.
>       - **Fig. 2:** The line represents the different choices along the Rate-Distortion curve which may be chosen by a user compressing with our algorithm. As described in the paper, the flexibility to work at any desired rate is an advantage of our compression method, while also existing in several classic approaches such as JPEG.
>       - **Fig. 4:** We will modify the diagram to add detail and improve its quality.
>       - **Fig. 5:** The results exhibit lower quality on the right due to the much lower compression rate, as indicated by the BPP (bits per pixel) values on the images.
>       - **Fig. 6:** The  results on the right lose detail due to the lack of a text-prompt in generation, causing less specificity in the images generated for the active acquisition phase and thus causing image quality to drop.
>
> [1] Song, Yang, et al. "Score-based generative modeling through stochastic differential equations." arXiv preprint arXiv:2011.13456 (2020).
>
> [2] Kingma, Diederik, et al. "Variational diffusion models." Advances in neural information processing systems 34 (2021): 21696-21707.

---

### Official Review · Reviewer_tEhF · 2024-11-02

**Soundness:** 2
**Presentation:** 1
**Contribution:** 2
**Rating:** 3
**Confidence:** 3

**Summary:**

This paper proposes a zero-shot image compression method that utilizes pre-trained diffusion models to compress images without further training. The progressive encoding strategy has a high computational cost, but allows balancing distortion and perception.

**Strengths:**

* The main strength is that the approach is *training-free*, leveraging pre-trained diffusion models for zero-shot compression.
* The progressive encoding strategy allows balancing rate-distortion-perception based on application.
* The quality of the results are competitive with approaches that require further training or bespoke architectures.

**Weaknesses:**

* The main weakness is that the high computational costs associated with this approach are not well-discussed, and the high-level theoretical framework, justifying the strategy, is lacking in presentation.
   * The results do not meet the standard of ICLR, as they are not state-of-the-art while incurring a presumably high computational cost. The performance (time/params per model) is not clearly presented, e.g. with tables and figures.
* The paper has many steps building on existing works (AdaSense) which are not well-known in the community, making it difficult to follow or understand the high-level theory underpinning this work. I found the pseudocode in part C of the supplementary material to be more informative for the general approach. In general, several parts of the methodology could be presented at a higher level, rather than reading like dependent steps of a complicated recipe.
* Discussion on some important related research is missing, such as Gao et al., NeurIPS 2022, "Flexible Neural Image Compression via Code Editing" and Frequency Aware Transformer, Li et al., (ICLR 2024).

**Questions:**

* Can you show a comparative table with performance data, such as the encoding/decoding time/#pretrained model params/with along with quality measures /R-D/B-D rates? If this is better than expected, I would happily change my opinion.
* How does this compare with Gao et al., NeurIPS 2022, "Flexible Neural Image Compression via Code Editing" that has a single decoder and can be adopted on existing pre-trained models and Li et al., ICLR 2024 "Frequency-Aware Transformer" (state-of-the-art in learned compression)?

---

> ### Author Response · Authors · 2024-11-20
>
> - **High Computational Cost:** As mentioned, PSC requires many NFEs to generate the multitudes of samples required at each step. More specifically, we used an order of 10,000 NFEs for compressing each image (depending on the required rate of course, as our method is progressive). We stress that this is the first introduction of such a posterior-sampler based compression method, and we expect that major speed-ups could be made in the near future, with the improvements of diffusion models and posterior samplers (such as flow-based methods) in general, and with specific algorithmic ideas that fit the architecture of the proposed algorithm.
>  - **ICLR Standards:** Our paper presents an original formulation for using existing diffusion models for image compression; We stress that there are no similar posterior sampling-based compression methods existing in the literature. The introduced method is SOTA - it performs competitively to HiFiC [1], a leading training-based neural compression method. Furthermore, the proposed scheme is very flexible, offering a progressive compression that other learning-based methods do not serve. True, the proposed algorithm is of high complexity, as described above, and we believe that future work will seek and find ways to speed it up substantially. In summary, we believe our work meets the standards of ICLR and beyond.
>  - **Theoretical Framework:** We will revise the paper by better explaining the uncertainty reduction framework that justifies our method.
>  - **Complicated Exposition:** We thank the reviewer for this suggestion, and we will simplify our exposition, specifically of AdaSense.
>  - **Related Research:** We will refer to these works in our revision, thank you for bringing these works to our attention. The cited works are neural compression methods, and are trained directly for image compression, unlike our zero-shot method. We note that while [2] is flexible in terms of compression rate, similar to our method, it also requires hundreds or thousands of computationally costly NFEs to achieve its flexibility.
>
> [1] Mentzer, Fabian, et al. "High-fidelity generative image compression." Advances in Neural Information Processing Systems 33 (2020): 11913-11924.
>
> [2] Gao, Chenjian, et al. "Flexible neural image compression via code editing." Advances in Neural Information Processing Systems 35 (2022): 12184-12196.

---

### Official Review · Reviewer_z4bx · 2024-11-03

**Soundness:** 2
**Presentation:** 2
**Contribution:** 2
**Rating:** 3
**Confidence:** 3

**Summary:**

The paper proposes a ``progressive'' scheme known as Posterior Sampling-based Compression (PSC) for image compression utilizing diffusion priors. Notably, the method operates in a zero-shot manner, meaning it requires no additional training. The image compression process, represented as \( y = Hx \), constructs the matrix \( H \) row-by-row with the assumption that an initial matrix and quantized measurements are shared between the encoder and decoder. This design, combined with the Adasense algorithm, ensures that the matrix \( H \) can be identically generated on both the encoder and decoder sides, thereby negating the need for communication of side information. The Adasense algorithm leverages a diffusion posterior sampler to draw multiple samples, subsequently computing PCA to estimate the top principal component of the posterior distribution's covariance \( p(x \mid H_{0:k}, Q(y_{0:k})) \).

In summary, while the paper introduces an interesting approach, substantial improvements are needed in terms of structure and presentation. Additionally, the technical novelty of the work appears limited, as the main components are not original contributions.

**Strengths:**

- The proposed method demonstrates good performance.
- The empirical results are reasonable.
- The paper extends the discussion to include latent diffusion models enhanced with text conditioning, which improves compression performance.

**Weaknesses:**

- The writing in the paper could be significantly improved to make the contributions clearer. The primary component of the algorithm, Adasense, is not a novel contribution of this work.
- The abstract claims the method is practical and simple, but there is no analysis or data provided to quantify the number of function evaluations required to achieve the reported results. For example, posterior sampling methods like DDRM or PiGDM typically involve at least 100 NFEs. Depending on the number of posterior samples used in line 1 of Algorithm 1, the cost could be as high as \( s \times 100 \) NFEs.
- The paper does not provide sufficient clarity on how the active acquisition strategy helps reduce uncertainty.
- Two main components of the approach---posterior sampling methods (e.g., DDRM or PiGDM) and the Adasense algorithm---are not novel contributions of this work, limiting its originality.
-  While the paper mentions ``high computational cost,'' it does not provide any quantification or comparative analysis with other algorithms.
- The rationale for not considering non-linear measurements is unclear, as posterior samplers for non-linear inverse problems exist in the literature.

**Questions:**

- Why is PiGDM considered superior to DDRM for the compression task?
- What is the performance gap between the proposed method and neural compression-based methods?
- PSC is stated to be limited to linear measurements due to the posterior sampler. However, samplers such as DPS can handle both linear and non-linear inverse problems. Did the authors explore this option?
- Did the authors attempt to use more efficient generative models, such as consistency models, to reduce the number of NFEs?

---

> ### Author Response · Authors · 2024-11-20
>
> - **Writing Quality:** We will make a sincere effort to make the contributions clearer.
>  - **Required NFEs:** We use 25 NFEs to generate samples with DDRM (as specified in the supplementary material). As for the overall complexity, we discuss it hereafter. We will make these details clear in a revised version.
>  - **Reducing Uncertainty:** We define the uncertainty as the MSE attained by the linear minimum-MSE (MMSE) predictor of x based on the measurements y. With this definition, the uncertainty directly correlates to the posterior distribution’s covariance, and thus a closed-form solution for the optimal linear measurement exists. By acquiring additional measurements, or in our case increasing the size of the compressed representation, we can decrease this measurement of uncertainty in the final restored image.
>  - **Novelty of Method:** Indeed, neither the posterior samplers used or active acquisition methods are contributions of our work. Nevertheless, we believe that the use of these pre-existing tools (which also include the pretrained diffusion models in our case) for new domains is both useful and original. We note that we are not aware of any other work employing posterior sampling of any kind, regardless of the methods, for image compression.
>  - **High Computational Cost:** As mentioned, PSC requires many NFEs to generate the multitudes of samples required at each step. More specifically, we used an order of 10,000 NFEs for compressing each image (depending on the required rate of course, as our method is progressive). We stress that this is the first introduction of such a posterior-sampler based compression method, and we expect that major speed-ups could be made in the near future, with the improvements of diffusion models and posterior samplers (such as flow-based methods) in general, and with specific algorithmic ideas that fit the architecture of the proposed algorithm.
>  - **Non-Linear Measurements:** while diffusion-based non-linear solvers exist, they have yet to be employed for an active acquisition scheme (such as AdaSense). This is probably because finding the optimal non-linear measurements even for L2 error is non-trivial, unlike the linear case. While developing such an active acquisition scheme would be of high interest, we believe it to be out of the scope of our work, which attempts to use the existing active acquisition tools for image compression.
>  - **PiGDM:** We find that restoring the compressed representation with PiGDM yields images with better perceptual quality. We highlight that this does not necessarily make them better, as their distortion is greater than the restorations produced by DDRM. This tradeoff is expected from different restoration algorithms according to the Perception Distortion Tradeoff [1], and we leave it to the user to choose what restoration algorithm to use to restore the compressed representation. Also, we note that PiGDM was not used during the active acquisition phase of PSC, but only during the final restoration.
>  - **Neural Compression:** We have compared PSC to one notable neural compression method, HiFiC [2], and we reach slightly better performance. Yet, as a zero-shot method that has never seen the task of image compression in training, we do not expect our method to be SOTA compared to all training-based neural compression methods. We should mention that in our experiments, we ran our method in its simplest form, without additional improvements, such as better entropy coding, better optimization of the projection directions, etc..
>  - **More Efficient Models:** To the best of our knowledge, methods such as consistency models (CM) are not designed for posterior sampling of the form we require in the proposed compression algorithm. Adjusting CM to accommodate this task will need many NFE’s as it will necessarily rely on an iterative evaluation of the network. The exploration of such faster sampling methods is an interesting future work.
>
>
> [1] Blau, Yochai, and Tomer Michaeli. "The perception-distortion tradeoff." Proceedings of the IEEE conference on computer vision and pattern recognition. 2018.
>
> [2] Mentzer, Fabian, et al. "High-fidelity generative image compression." Advances in Neural Information Processing Systems 33 (2020): 11913-11924.

---

### Author Response · Authors · 2024-11-27

We thank the reviewers for their comments. We have updated our paper according to some of their suggestions.
* We attempted to simplify the explanations of our use of "uncertainty" and the reasons for using linear measurements for active aquisition.
* We have highlighted the computational cost of this preliminary version of our proposed methods, and expanded the related discussion.
* We have expanded the description of the figures in the main text.
* A comparison to ELIC[1] and IPIC[2] has been added to 2, showing a comparison to more recent neural compression algorithms. While we do not claim to be SOTA, we believe the new results strengthens our conclusion that our method is competitive, espectially as a zero-shot method. We have not been able to add comparisons to all methods requested by the reviewers due to the lack of publicly available implementations and the short amount of time for the rebuttal.

[1] He, Dailan, et al. "Elic: Efficient learned image compression with unevenly grouped space-channel contextual adaptive coding." Proceedings of the IEEE/CVF Conference on Computer Vision and Pattern Recognition. 2022.

[2] Xu, Tongda, et al. "Idempotence and perceptual image compression." arXiv preprint arXiv:2401.08920 (2024).

---

### Note · Authors · 2025-01-16

**Comment:**

We have decided to withdraw our paper, we thank the AC and the reviewers.

**Withdrawal Confirmation:**

I have read and agree with the venue's withdrawal policy on behalf of myself and my co-authors.